# Survival of mineral-bound peptides into the Miocene

Beatrice Demarchi[1]*, Meaghan Mackie[2,3], Zhiheng Li[4], Tao Deng[4], Matthew J Collins[2,5], Julia Clarke[6]

[1]Department of Life Sciences and Systems Biology, University of Turin, Torino, Italy; [2]The Globe Institute, Faculty of Health and Medical Sciences, University of Copenhagen, Copenhagen, Denmark; [3]The Novo Nordisk Foundation Center for Protein Research, Faculty of Health and Medical Sciences, University of Copenhagen, Copenhagen, Denmark; [4]Institute of Vertebrate Paleontology and Paleoanthropology, Chinese Academy of Sciences, Beijing, China; [5]McDonald Institute for Archaeological Research, University of Cambridge, Cambridge, United Kingdom; [6]Department of Geological Sciences, The University of Texas at Austin, Austin, United States

**Abstract** Previously, we showed that authentic peptide sequences could be obtained from 3.8-Ma-old ostrich eggshell (OES) from the site of Laetoli, Tanzania (Demarchi et al., 2016). Here, we show that the same sequences survive in a >6.5 Ma OES recovered from a palaeosteppe setting in northwestern China. The eggshell is thicker than those observed in extant species and consistent with the Liushu *Struthio* sp. ootaxon. These findings push the preservation of ancient proteins back to the Miocene and highlight their potential for paleontology, paleoecology, and evolutionary biology.

## Editor's evaluation

This fundamental study substantially pushes the known preservation of protein sequences bound to mineral surfaces. The successful recovery of these sequences from late Miocene fossil eggshell also has important implications for taxonomic classification. This solid work encourages future paleoproteomic research on paleontological remains from deep antiquity and across various taxa. The paper will be of great interest to a wide range of paleoscientists.

*For correspondence:
beatrice.demarchi@unito.it

**Competing interest:** The authors declare that no competing interests exist.

## Introduction

The oldest authenticated peptide sequences to date were reported in 2016 from 3.8-Ma-old ostrich eggshell (OES) from the site of Laetoli, Tanzania (*Demarchi et al., 2016*). This finding had great scientific impact since it integrated computational chemistry (molecular dynamics simulations) as well as experimental data to propose a mechanism of preservation, concluding that mineral binding ensures the survival of protein sequences. Importantly, this study demonstrated that peptide-bound amino acids could survive into deep time even in hot environments. The effect of temperature on the kinetics of protein diagenesis has been described by several authors, both on the basis of actualistic experiments and of the quantification of the extent of degradation in ancient samples of known ages (*Crisp et al., 2013*; *Demarchi et al., 2013*; *Hendy et al., 2012*; *Johnson et al., 1997*; *Kaufman, 2006*; *Kaufman, 2003*; *Schroeder and Bada, 1976*; *Wehmiller, 2013*; *Wehmiller, 1977*; *Wehmiller and Belknap, 1978*). This discovery has fuelled the analysis of ancient proteins from other mineral matrices, namely tooth enamel (*Cappellini et al., 2019*; *Welker et al., 2020*; *Welker et al., 2019*) in order to reconstruct phylogenies.

**Figure 1.** Map showing the location of the fossil eggshell site.

The recovery of peptides has so far been limited to biomineral samples of Plio-Pleistocene age: our attempts at retrieving intact peptides from Cretaceous eggshell were unsuccessful, despite the fact that a genuine intracrystalline fraction of amino acids was preserved in the same sample (*Saitta et al., 2020*). Here, we show that the same peptide sequences that exhibit strong binding to the calcite surface—and that were recovered from the Laetoli OES—also persist into the late Miocene, in a >6.5-Ma-old eggshell sample from the Linxia Basin, northeastern Tibetan Plateau, China, Liushu Formation.

## Results and discussion

The OES (IVPP V26107) was recovered from an area between the towns of Xinji and Songming, in Hezheng County, close to the border with Guanghe County, from mudstone facies of the Liushu Formation (*Figure 1*).

Both eggshell and skeletal remains are previously known from the Late Miocene Liushu Formation, Linxia Basin, of Gansu Province, where the mean annual temperature is ~11°C (*Hou et al., 2005*; *Liu et al., 2016*; *Li et al., 2021*; *Wang, 2008*). Age control on the highly fossiliferous units of the Lishu Formation is good (*Deng et al., 2019*; *Deng et al., 2013*; *Zhang et al., 2012*), with detailed correlations across China as well as with Neogene deposits globally. These units have been assessed via new

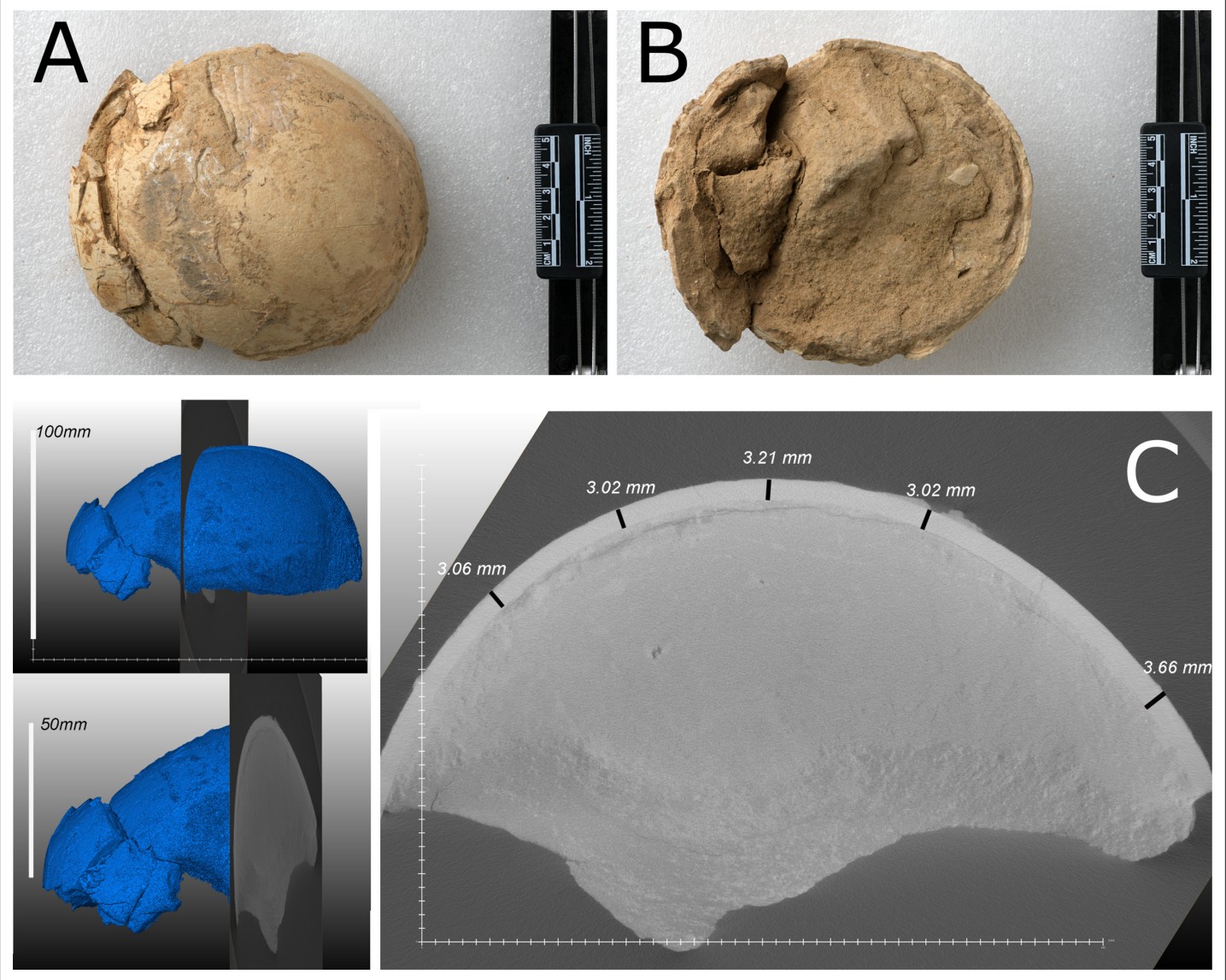

**Figure 2.** Eggshell specimen (IVPP V26107): photographs (**A, B**) and CT scan (**C**).

magnetostratigraphic data; most of the *Hipparion* fauna is estimated to be earlier than 6.4–6.5 Ma, with a lower Liushu transition to a *Platybelodon* fauna yielding age estimates of 11.1–12.5 Ma (***Zhang et al., 2012***). These dates agree with the prior estimates from biostratigraphic and magnetostratigraphic data (***Deng et al., 2019***; ***Deng et al., 2013***). The recovery site is closer to the southern part of the basin near the Heilinding section of ***Zhang et al., 2012***, which had an estimated minimum age of 6.4–6.5 Ma (Chron 3An). In contrast, the candidate stratotype section, Guoniguo, that exposes lower parts of the Liushu, is north of the recovery area (***Deng et al., 2019***; ***Zhang et al., 2012***).

Ostrich (*Struthio*) remains to have a long history of recovery from the late Miocene of northwest China (***Buffetaut and Angst, 2021***; ***Hou et al., 2005***; ***Li et al., 2021***; ***Lowe, 1931***; ***Mikhailov and Zelenkov, 2020***). The first reported ostrich fossils from China were from units of the '*Hipparion* Clay' (Red Clay) dated between 6.54 and 7.18 Ma (***Lowe, 1931***; ***Zhu et al., 2008***). ***Mikhailov and Zelenkov, 2020*** concluded that the ***Wang, 2008*** Liushu taxon is referable to *Struthio* and not presently supported as a distinct but related genus; it is proposed to be from a taxon larger than extant *Struthio* species with a thickness of ~2.4 mm consistent with the eggshell sampled here (***Figure 2C***).

Given the extreme antiquity of the samples, protein extraction was performed in an ultra-clean facility at the University of Copenhagen in order to minimize any chance of contamination, following

**Table 1.** Peptide sequences identified in sample IVPP V26107.

| Scan | Precursor mass | z | m/z | Peptide sequence | Peptide mass | Error (ppm) | Peptide –10lgP |
|------|----------------|---|-------|------------------|--------------|-------------|----------------|
| 5401 | 1107.472 | 2 | 555.246 | ALDDDDYPKG | 1107.472 | 5.7 | 27.24 |
| 5268 | 1050.457 | 2 | 526.236 | ALDDDDYPK | 1050.451 | 6.1 | 24.83 |
| 5051 | 1194.517 | 2 | 598.266 | SALDDDDYPKG | 1194.504 | 10.9 | 21.11 |
| 4984 | 1137.494 | 2 | 569.754 | SALDDDDYPK | 1137.483 | 10.1 | 29.94 |
| 4727 | 1137.494 | 2 | 569.755 | SALDDDDYPK | 1137.483 | 9.7 | 16.76 |
| 4565 | 1107.483 | 2 | 554.749 | ALDDDDYPKG | 1107.472 | 9.8 | 32.57 |
| 4565 | 1107.484 | 2 | 554.749 | ALDDDDYPKG | 1107.472 | 10.6 | 28.7 |
| 4443 | 1050.459 | 2 | 526.237 | ALDDDDYPK | 1050.451 | 7.6 | 17.62 |
| 4313 | 1050.461 | 2 | 526.235 | ALDDDDYPK | 1050.451 | 9.9 | 15.13 |
| 4199 | 1050.458 | 2 | 526.236 | ALDDDDYPK | 1050.451 | 7.2 | 16.35 |
| 4037 | 1036.442 | 2 | 519.230 | LDDDDYPKG | 1036.435 | 7.2 | 16.57 |
| 2724 | 1164.539 | 2 | 583.277 | LDDDDYPKGK | 1164.53 | 8.2 | 30.91 |
| 2502 | 866.335 | 2 | 434.176 | DDDDYPK | 866.335 | 6.9 | 18.46 |
| 2176 | 866.335 | 2 | 434.175 | DDDDYPK | 866.335 | 7.1 | 21.18 |
| 2026 | 866.338 | 2 | 434.176 | DDDDYPK | 866.329 | 10.1 | 19.07 |
| 1396 | 1051.454 | 2 | 526.734 | DDDDYPKGK | 1051.446 | 7.8 | 28.62 |
| 1138 | 936.427 | 2 | 469.221 | DDDYPKGK | 936.419 | 8.4 | 27.8 |
| 1046 | 1188.513 | 3 | 397.178 | DDDDYPKGKH | 1188.505 | 7.2 | 30.65 |
| 1042 | 821.398 | 2 | 411.706 | DDYPKGK | 821.392 | 7.6 | 17.39 |

*Hendy et al., 2018*. Furthermore, OES powders were bleached extensively to isolate the intracrystalline fraction only (as described by *Demarchi et al., 2016*), samples were analysed by LC-MS/MS using a new LC column, and six blanks (analytical and procedural) were included in the run, flanking the sample. Data analysis was equally cautious: raw tandem mass spectrometry data were used to reconstruct potential peptide sequences starting from raw product ion spectra; only de novo peptides with an ALC (Average Local Confidence) score ≥80% were considered. This is the most stringent threshold that can be applied, and it signifies that all potential peptide sequences with lower confidence are discarded. The de novo peptides were searched against the Uniprot/Swissprot database (containing 565,254 manually annotated and reviewed protein sequences). A database of common laboratory contaminants (common Repository of Adventitious Protein [cRAP]) was included in the search. Eleven unique peptide sequences were found to match the sequence of struthiocalcin-1 (*Table 1*), all containing the typical Asp-rich motif 'DDDD' (*Figures 3 and 4*), which had been shown in our previous paper to be the mineral-binding peptide belonging to the sequence of struthiocalcin-1 (SCA-1). Annotated tandem mass spectra are shown in *Figure 3* (ALDDDDYPKG) and *Figure 3—figure supplement 1* (ALDDDYPK), *Figure 3—figure supplement 2* (SALDDDDYPKG), *Figure 3—figure supplement 3* (DDDDYPKGKH), *Figure 3—figure supplement 4* (LDDDDYPKGK), *Figure 3—figure supplement 5* (SALDDDDYPK), *Figure 3—figure supplement 6* (DDDDYPKGK), *Figure 3—figure supplement 7* (LDDDDYPKG), *Figure 3—figure supplement 8* (DDYPKGK), *Figure 3—figure supplement 9* (DDDDYPK), and *Figure 3—figure supplement 10* (DDYPKGK).

## Conclusions

We present the first evidence for peptide survival into the Miocene, confirming our previous (Pliocene) data and providing further support to the mechanism of preservation based on the binding of Asx-rich peptides to calcite surfaces (*Demarchi et al., 2016*). The sequence recovered from the Chinese specimen is identical to the peptides found in the 3.8 Ma Laetoli OES, thus also supporting the attribution of the Liushu ootaxon to genus *Struthio*. While the four Asp residues are conserved

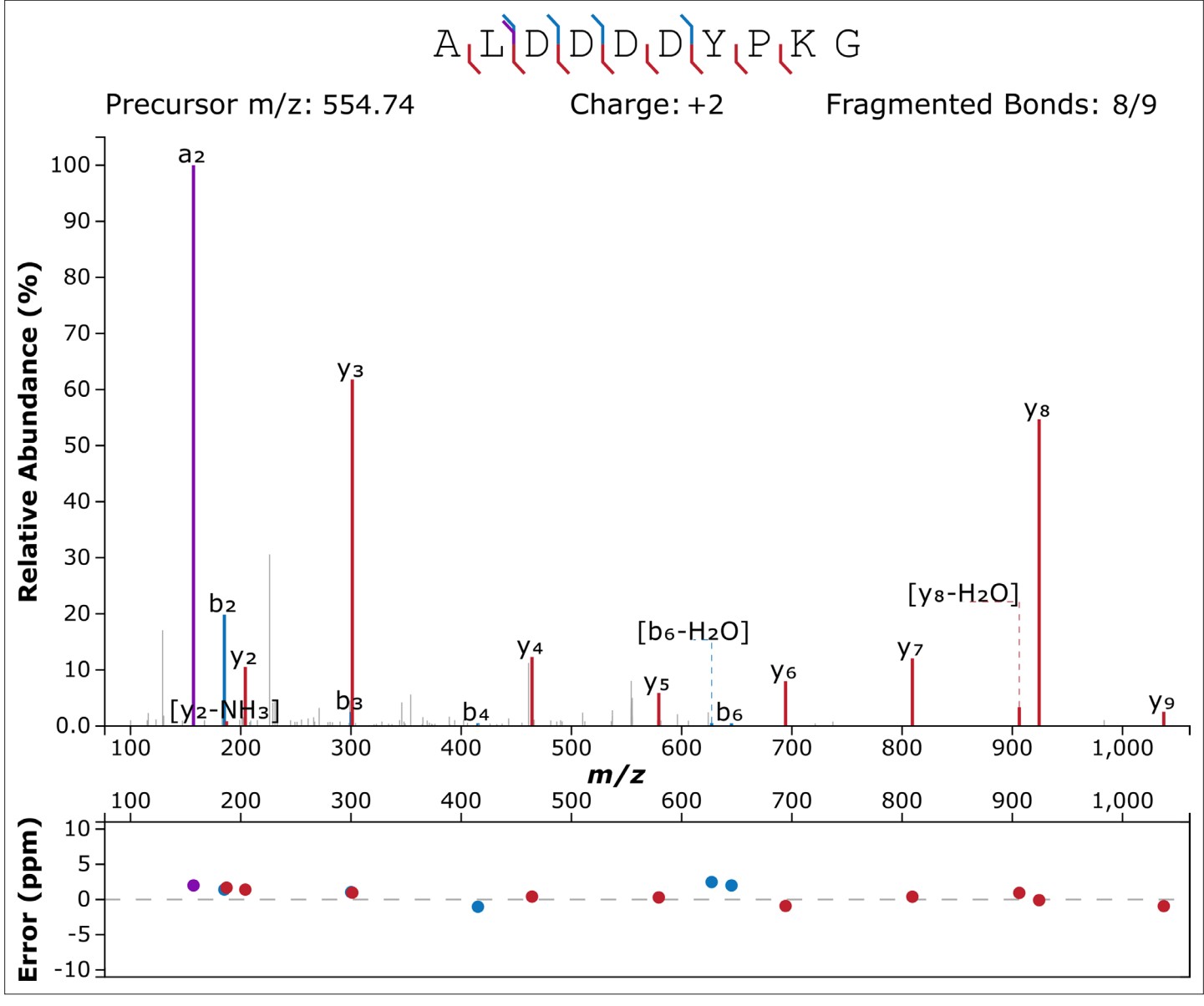

**Figure 3.** Annotated product ion spectrum of peptide ALDDDDYPKG, m/z 554.749, −10lgP=32.57. Figure created using http://www.interactivepeptidespectralannotator.com/PeptideAnnotator.html (*Brademan et al., 2019*).

The online version of this article includes the following figure supplement(s) for figure 3:

**Figure supplement 1.** Annotated product ion spectrum, ALDDDDYPK, m/z 526.236, −10lgP=24.83.

**Figure supplement 2.** Annotated product ion spectrum, SALDDDDYPKG, m/z 598.266, −10lgP=21.11.

**Figure supplement 3.** Annotated product ion spectrum, DDDDYPKGKH, m/z 397.178, −10lgP=30.65.

**Figure supplement 4.** Annotated product ion spectrum, LDDDDYPKGK, m/z 583.277, −10lgP=30.91.

**Figure supplement 5.** Annotated product ion spectrum, SALDDDDYPK, m/z 569.754, −10lgP=29.94.

**Figure supplement 6.** Annotated product ion spectrum, DDDDYPKGK, m/z=526.734, −10lgP=28.62.

**Figure supplement 7.** Annotated product ion spectrum, LDDDDYPKG, m/z=519.23, −10lgP=16.57.

**Figure supplement 8.** Annotated product ion spectrum, DDDYPKGK, m/z=469.221, −10lgP=27.8.

**Figure supplement 9.** Annotated product ion spectrum, DDDDYPK, m/z=434.174, −10lgP=21.18.

**Figure supplement 10.** Annotated product ion spectrum, DDYPKGK, m/z=411.706, −10lgP=17.39.

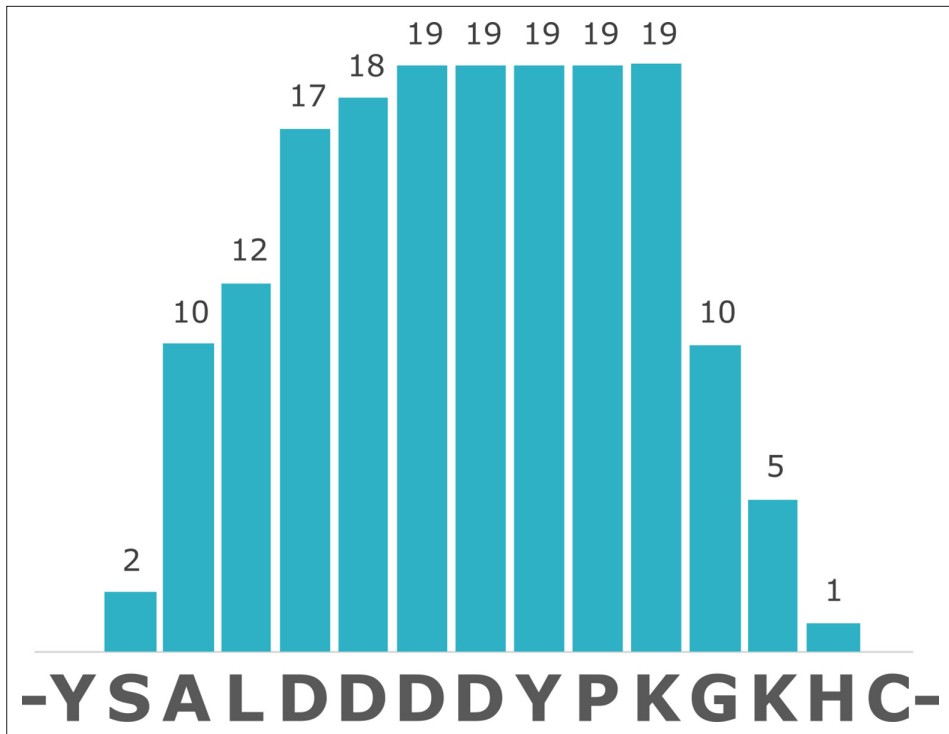

**Figure 4.** Coverage of struthiocalcin-1 (SCA-1) in Miocene OES specimen IVPP V26107. Numbers indicate spectrum matches for each position in the sequence.

across several avian taxa, in some species, including other ratites, Asp can be substituted by Glu and some of the flanking residues are also variable (*Demarchi et al., 2022*, fig. S1); therefore, variability within this sequence could be informative for evolutionary relationships between extinct and extant taxa. The absolute age of the Chinese sample is greater than that of the African material (*Demarchi et al., 2016*), but, given the latitudinal differences between the two sites, it is likely to have experienced lower temperatures throughout its burial history. This suggests that sequences of even greater antiquity may be recovered from biominerals harboring a closed system of proteins, particularly from sites in cold environments (e.g., high altitude and/or latitude).

## Materials and methods
### Sample preparation and analysis
A subsample of OES specimen IVPP V26107 was prepared in the ultra-clean facility at the University of Copenhagen, following the protocol of *Demarchi et al., 2016*; *Demarchi et al., 2022* and omitting the digestion step. In brief, the fragment was powdered, bleached for 72 hr (NaOCl, 15% w/v) and demineralized in cold weak hydrochloric acid (0.6 M HCl). The acid was added in 200 µl increments until it stopped reacting, for a total volume of 1400 µl 0.6 M HCl. The extracts were exchanged in ammonium bicarbonate buffer (pH=7.8) using 3 kDa MWCO ultrafilters and the peptides were purified and concentrated using C18 Stage Tips (*Cappellini et al., 2019*; *Rappsilber et al., 2007*).

Prepared StageTips of the procedural blank and sample were eluted with 30 µl of 40% acetonitrile (ACN) 0.1% formic acid into a 96-well plate prior to LC-MS/MS analysis. To remove the ACN, the plate was vacuum centrifuged until approximately 5 µl remained. Samples were then resuspended with 6 µl of 0.1% trifluoroacetic acid (TFA) 5% ACN. Based on protein concentration results at 205 nm (Nano-Drop, Thermo Fisher Scientific), 5 µl of each sample and procedural blank was then separated over a 77 min gradient by an EASY-nLC 1200 (Proxeon, Odense, Denmark) attached to a Q-Exactive HF-X mass spectrometer (Thermo Fisher Scientific, Germany) using a 15 cm column. The column (75 µm inner diameter) was made in-house, laser pulled, and packed with 1.9 µm C18 beads (Dr. Maisch, Germany). Parameters were the same as those already published for historical samples (*Mackie et al.,*

*2018*). In short, MS1: 120k resolution, maximum injection time (IT) 25 ms, scan target 3E6. MS2: 60k resolution, top 10 mode, maximum IT 118 ms, minimum scan target 3E3, normalized collision energy of 28, dynamic exclusion 20 s, and isolation window of 1.2 m/z. Wash-blanks consisting of 0.1% TFA 5% ACN were also run in order to hinder cross-contamination. The LC-MS/MS run included, in this order: two wash blanks, one procedural blank, one wash blank, the OES sample, and two wash blanks. The data sets have been deposited to the ProteomeXchange Consortium via the Proteomics Identifications Database (PRIDE) partner repository with the identifier PXD035872.

## Data analysis
Bioinformatic analysis was carried out using PEAKS Studio 8.5 (Bioinformatics Solutions Inc; *Zhang et al., 2012*). The Uniprot_swissprot database (downloaded 11/08/2021) was used for carrying out the searches and common contaminants were included (cRAP: http://www.thegpm.org/crap/). No enzyme was specified for the digestion and the tolerance was set to 10 ppm on the precursor and 0.05 Da on the fragments. The thresholds for peptide and protein identification were set as follows: peptide score −10lgP≥15, protein score −10lgP≥20, and de novo sequences scores (ALC%)≥80.

## Acknowledgements

BD's work was supported by a "Young Researchers - Rita Levi Montalcini" grant from the Italian Ministry of University and Research. MM and MJC were supported by the Danish National Research Foundation Award PROTEIOS (DNRF128). JC acknowledges the Alexander von Humboldt Foundation and the Jackson School of Geosciences and ZL and DT the Chinese National Science Foundation. The authors thank Prof. Jesper Velgaard Olsen at the Novo Nordisk Center for Protein Research for providing access and resources, which were also funded in part by a donation from the Novo Nordisk Foundation (Grant no. NNF14CC0001). For the purpose of open access, MJC is part of the Rights Retention Pilot at the University of Cambridge and has applied a Creative Commons Attribution (CC BY) licence to any Author Accepted Manuscript version arising from this submission. The authors wish to thank the reviewers and editors for their feedback on this manuscript.

## Additional information

### Funding

| Funder | Grant reference number | Author |
| --- | --- | --- |
| Ministry of University and Research - Italy | Young Researchers - Rita Levi Montalcini | Beatrice Demarchi |
| Danish National Research Foundation | PROTEIOS (DNRF128) | Meaghan Mackie Matthew J Collins |
| Alexander von Humboldt Foundation | | Julia Clarke |
| University of Texas at Austin | | Julia Clarke |
| National Natural Science Foundation of China | | Zhiheng Li |

The funders had no role in study design, data collection and interpretation, or the decision to submit the work for publication.

### Author contributions
Beatrice Demarchi, Conceptualization, Data curation, Formal analysis, Funding acquisition, Investigation, Visualization, Methodology, Writing – original draft, Project administration, Writing – review and editing; Meaghan Mackie, Formal analysis, Methodology, Writing – review and editing; Zhiheng Li, Formal analysis, Visualization, Writing – review and editing; Tao Deng, Resources, Investigation, Methodology, Writing – review and editing; Matthew J Collins, Conceptualization, Resources, Formal

analysis, Writing – original draft, Project administration, Writing – review and editing; Julia Clarke, Conceptualization, Investigation, Methodology, Writing – original draft, Writing – review and editing

### Author ORCIDs
Beatrice Demarchi http://orcid.org/0000-0002-8398-4409

### Decision letter and Author response
Decision letter https://doi.org/10.7554/eLife.82849.sa1
Author response https://doi.org/10.7554/eLife.82849.sa2

## Additional files

### Supplementary files
• Transparent reporting form

### Data availability
Tandem mass spectra supporting peptide sequence identification are reported in Figure 3 and Figure 3 - Supplement 1 to 10. Raw mass spectrometry data and results of bioinformatics analysis are available via ProteomeXchange with identifier PXD035872.

The following dataset was generated:

| Author(s) | Year | Dataset title | Dataset URL | Database and Identifier |
| --- | --- | --- | --- | --- |
| Demarchi B | 2022 | Ostrich eggshell peptides survival into the Miocene (6-9 Ma) | https://www.ebi.ac.uk/pride/archive/projects/PXD035872 | PRIDE, PXD035872 |

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
