## [Editor Report]

This fundamental study substantially pushes the known preservation of protein sequences bound to mineral surfaces. The successful recovery of these sequences from late Miocene fossil eggshell also has important implications for taxonomic classification. This solid work encourages future paleoproteomic research on paleontological remains from deep antiquity and across various taxa. The paper will be of great interest to a wide range of paleoscientists.

---

## [Decision Letter]

**Decision letter after peer review:**

Thank you for submitting your article "Survival of mineral-bound peptides into the Miocene" for consideration by *eLife*. Your article has been reviewed by 2 peer reviewers, and the evaluation has been overseen by a Reviewing Editor and George Perry as the Senior Editor. The reviewers have opted to remain anonymous.

1) Both reviewers found merit in your work and appreciated the significance of it for the field of paleoproteomics. While your successful identification and recovery of authentic peptide sequences from a late Miocene fossil eggshell pushes the known time range of ancient protein preservation, both reviewers questioned the relevance of the reported thermal age owing to its inherent limitations. Both reviewers highlighted that thermal age provides unreliable estimates, erring on the generous side. Because of this limitation, and the fact that there are more reliable chronometric dates for the analyzed eggshell sample, the relevance, and validity of the reported thermal age is unwarranted. As such, both reviewers recommended that the reported thermal age be omitted from the paper altogether, owing to the inherent limitations of it and the availability of more accurate geological dates for your sample.

2) Reviewer #2 strongly recommended that your MS data be reported, which I agree with.

Both reviewers have provided additional constructive comments, which I encourage you to address in your revision.

*Reviewer #1 (Recommendations for the authors):*

"The oldest authenticated peptide sequences to date were reported in 2016 from 3.8 Ma old ostrich eggshell (OES) from the site of Laetoli, Tanzania (Demarchi et al., 2016), with a thermal age of the peptides estimated to be equivalent to ~16 Ma at a constant 10°C."

Please remove all reference to thermal age from the entire manuscript. Thermal age isn't real and its usage obscures patterns of preservation/protein breakdown. Also, it artificially inflates the age of samples that are already old. The previously discussed 3.8 Ma ostrich egg shells are actually 3.8 Ma and not 16 Ma. The arbitrariness of thermal age is especially evident in the last paragraph before the acknowledgements considering local temperature across the entire 6.4-6.5 Ma cannot be known and the thermal age calculation is described as 'crude'. Additionally, this sample is actually 2.6-2.7 Ma older than the previously characterized eggshells regardless of whatever the thermal age model outputs.

"25 sequences were found with an ALC (Average Local Confidence) score {greater than or equal to} 80%. This is the most stringent threshold that can be applied" If this was only from the de novo search, please include a table listing the 25 de novo peptides. It may be helpful to indicate if they match the database search as well.

"individual annotated tandem mass spectra are shown in Figure 3 and Figure 3—figure supplement 1, Figure 3—figure supplement 2, Figure 3—figure supplement 3, Figure 3—figure supplement 4, Figure 3—figure supplement 5, Figure 3—figure supplement 6, Figure 3—figure supplement 7, Figure 3—figure supplement 8, Figure 3—figure supplement 9, Figure 3—figure supplement 10, Figure 3—figure supplement 11" Should it be Figure 4? Also, please list each peptide sequence here as well for ease of connecting to the supplemental figures.

Figure 4: Please include a table in the main manuscript with the 12 peptides. I appreciate that they are annotated in the supplement, but there is no reason to not include them. Alternatively, put the aligned peptides in figure 4 in place of the gray circles. I don't know if it is possible with the software used, but please try and make this figure colorblind accessible. Red and green together are not great. Please also include details on how this figure was produced.

"demineralised in just enough cold weak hydrochloric acid" Please quantify how much acid this is.

*Reviewer #2 (Recommendations for the authors):*

The weakest part of this manuscript is the inclusion of thermal age data, which is a metric I do not find reliable or informative in its own right, but moreover, its inclusion does not seem to add anything of value to the MS data that is presented. The thermal age estimates at the end of this paper constitute using multiple assumptions about ancient temperature and humidity values and their variability (e.g., from a "compilation of proxies") to return what the authors themselves describe as a "very crude" estimate that is nearly identical to the specimen's actual age. For what purpose? Whether "thermal age" as a metric is meant to be a proxy for determining the likelihood of the persistence of peptides (i.e., to assess that a fossil is good for analyses because it is likely less degraded than its age suggests) or to use for authentication of ancient peptides (i.e., to show that very old fossils are in fact 'thermally younger' than their geologic age because they've undergone less heating and therefore the preservation of proteins in them beyond the hypothetical limit of preservation is not problematic) both this paper and the earlier paper giving a thermal age of 16 Ma to the 3.8 Ma Laetoli eggshell show that neither usage is valid. In fact, if thermal age were to be used in this paper as it has been used in the past, it would suggest that the peptides the authors report here represent contamination. Of course, I think that would be absurd, and that the annotated spectra the authors produced in a sterile environment, with no trace of contamination in the blanks, confirm the authenticity of their peptides. However, both things can't be true. If the peptides are real--and I think they are--then thermal age is not a useful metric.

I strongly recommend the authors simply report their MS data, which I think are extremely impactful and worthy of publication on their own.

Technical Comments:

1. It is unclear to me how the authors came to the specific age range of 6.5 Ma to 9 Ma. The text is clear that 6.5 Ma is the minimum age estimate for their recovery site, and they also state that previously reported ostrich fossils were dated between 6.54-7.18 Ma, and that Platybelodon had been dated from lower in the Liushu at 11.1 – 12.5 Ma. So where does the upper bound estimate come from? Normally I wouldn't nitpick this point, but as the authors claim to have the oldest recovered peptides, the rationale for that upper bound is important and should be justified in the text. Otherwise, the authors should drop the range and stick with the minimum estimate of 6.5 Ma.

2. Figure 4 is visually difficult to interpret. I strongly suggest that you change this figure to depict the letters signifying the residues as equally sized (to make them readable) and place proportionately sized bars above them (like a column chart) to show their relative coverage.

3. Materials and methods: What is "just enough" HCl? Please give a volume.

4. It's unclear to me why the authors chose to use a fragment error tolerance of 0.5 Da, given that 0.05 Da or less is standard for ancient protein LC-MS/MS data, and that data from the Q-Exactive is certainly high enough resolution to set a higher tolerance. Additionally, the annotated peptides all appear to be within the 0.05 Da range. Is this a typo? If so, please correct it. If not, I'm curious about how a low fragment error tolerance coupled with a large search space (since it's a no-digest search) impacts the FDR rates. Please report the %FDR represented by a peptide score of −10lgP {greater than or equal to} 30 for the ostrich data.

[Editors' note: further revisions were suggested prior to acceptance, as described below.]

Thank you for resubmitting your work entitled "Survival of mineral-bound peptides into the Miocene" for further consideration by *eLife*. Your revised article has been evaluated by George Perry (Senior Editor) and a Reviewing Editor.

The manuscript has been improved but there are some remaining issues that need to be addressed, as outlined below:

I [George Perry] am writing this on behalf of Yonathan Sahle, who has departed for remote, international research, but we corresponded before he left so that I could incorporate his feedback into this decision letter here and get this back to you more efficiently than otherwise.

Specifically, while you have addressed most of the reviewers' concerns, the maintained position regarding the relevance of the thermal age clashes with the strong recommendations by both reviewers (a position the reviewing editor agrees with).

The geological age of the specimens should always be used as the true age, given that the thermal age is sensitive to many factors and has been repeatedly shown to artificially inflate the true age of paleontological remains. In lines 36-43 of the revised manuscript, the thermal age estimate for the Laetoli (Tanzania) specimen is invoked, despite the very large discrepancy of >10 Ma between the true age (3.6 Ma) and the thermal age (~16 Ma) of the Tanzanian sample.

True temperature affects protein preservation and there are several works that have tried to quantify the extent of temperature-induced protein degradation. However, (a) there are several unknowns (estimates) that affect such assessments (e.g., past temperatures) , (b) more samples and contexts are required for a better understanding of these temperature-protein diagenesis relationships, and (c) regardless of the level of confidence of the estimated/modelled thermal age, its relevance for the current paper on ancient proteins from geologically dated OES from Linxia Basin in China is unclear (see also the disconnect in Lines 171-173).

If the thermal age data are omitted altogether, or limited to a very brief and cautious mention in results/discussion, then we would likely be ready to move forward favorably with the paper. But this remaining point is considered essential.

---

## [Author Response]

The reviewers have discussed their reviews with one another, and the Reviewing Editor has drafted this to help you prepare a revised submission.1) Both reviewers found merit in your work and appreciated the significance of it for the field of paleoproteomics. While your successful identification and recovery of authentic peptide sequences from a late Miocene fossil eggshell pushes the known time range of ancient protein preservation, both reviewers questioned the relevance of the reported thermal age owing to its inherent limitations. Both reviewers highlighted that thermal age provides unreliable estimates, erring on the generous side. Because of this limitation, and the fact that there are more reliable chronometric dates for the analyzed eggshell sample, the relevance, and validity of the reported thermal age is unwarranted. As such, both reviewers recommended that the reported thermal age be omitted from the paper altogether, owing to the inherent limitations of it and the availability of more accurate geological dates for your sample.

We thank the Reviewers and the Editors for their positive assessment of our manuscript. We agree that, in the present case, thermal ages are affected by significant errors (because of the lack of paleotemperature models for this area, and for ages that span all the way back to the Miocene), and therefore only "crude estimates" can be provided. As such, we removed the calculated thermal ages as requested. However, we wish to clarify that thermal ages were not used to provide a chronometric assessment of the samples (neither here nor in our previous paper), but rather as a “diagenesis” index, which allows us to compare the extent of degradation in samples from very different environments. Indeed, protein diagenesis is strongly affected by temperature (Crisp et al., 2013; Demarchi et al., 2013; Hendy et al., 2012; Johnson et al., 1997; Kaufman, 2006, 2003; Schroeder and Bada, 1976; Wehmiller, 2013, 1977; Wehmiller and Belknap, 1978). This effect cannot be ignored in palaeoproteomics – proteins extracted from identical samples from sites in the Tropics vs the Arctic will *not* show the same extent of preservation even if the samples were of exactly the same age and composition. In order to draw the reader's attention to this important concept we added the following sentence to the Introduction:

“The effect of temperature on the kinetics of protein diagenesis has been described by several authors, both on the basis of actualistic experiments and by quantifying the extent of degradation in ancient samples of known ages (Crisp et al., 2013; Demarchi et al., 2013; Hendy et al., 2012; Johnson et al., 1997; Kaufman, 2006, 2003; Schroeder and Bada, 1976; Wehmiller, 2013, 1977; Wehmiller and Belknap, 1978). By modelling the combined effect of temperature and time on protein diagenesis in a closed system, the peptides recovered from the Laetoli OES were estimated to be equivalent to ~16 Ma at a constant 10°C.”

2) Reviewer #2 strongly recommended that your MS data be reported, which I agree with.

In addition to the tandem mass spectra, we now also present a table with all the identified peptides.

Both reviewers have provided additional constructive comments, which I encourage you to address in your revision.Reviewer #1 (Recommendations for the authors):"The oldest authenticated peptide sequences to date were reported in 2016 from 3.8 Ma old ostrich eggshell (OES) from the site of Laetoli, Tanzania (Demarchi et al., 2016), with a thermal age of the peptides estimated to be equivalent to ~16 Ma at a constant 10°C."Please remove all reference to thermal age from the entire manuscript. Thermal age isn't real and its usage obscures patterns of preservation/protein breakdown. Also, it artificially inflates the age of samples that are already old. The previously discussed 3.8 Ma ostrich egg shells are actually 3.8 Ma and not 16 Ma. The arbitrariness of thermal age is especially evident in the last paragraph before the acknowledgements considering local temperature across the entire 6.4-6.5 Ma cannot be known and the thermal age calculation is described as 'crude'. Additionally, this sample is actually 2.6-2.7 Ma older than the previously characterized eggshells regardless of whatever the thermal age model outputs.

As requested, we have removed our crude estimate of thermal age from the text, as well as the whole final paragraph. However, we respectfully point out that theoretical kinetic models of racemization have been used for half a century to correlate sites of equivalent geological age but with different thermal histories and thus different degrees of protein degradation (e.g. Figure 6 of Wehmiller, J. F. 2013. United States Quaternary coastal sequences and molluscan racemization geochronology – What have they meant for each other over the past 45 years? Quaternary Geochronology, 16, 3–20).

Further, it is demonstrable that when eggshell experiences differences in temperature at a single site (e.g. over glacial/interglacial cycles), this results in corresponding fluctuations in rates of protein degradation (e.g. Figure 9 of Miller, G. H., Fogel, M. L., Magee, J. W., and Gagan, M. K. (2016). Disentangling the impacts of climate and human colonization on the flora and fauna of the Australian arid zone over the past 100 ka using stable isotopes in avian eggshell. Quaternary Science Reviews, 151, 27–57).

Protein degradation in closed systems is influenced by temperature (which thermal age attempts to normalize). Despite the greater antiquity of the samples reported here, our (admittedly crude) calculations suggest that their persistence is in fact not as remarkable as the sequences recovered from Africa. The take home message from the reviewer that “this sample is actually 2.6-2.7 Ma older than the previously characterized eggshells regardless…” therefore minimizes the significance of the African findings, whose thermal age predicted the persistence of sequences of greater antiquity in higher latitudes and consequently even greater antiquity at still higher altitudes and latitudes.

We have added the following sentence to the conclusions: The absolute age of the Chinese sample is greater than that of the African material (Demarchi et al., 2016), but it is estimated to have experienced lower temperatures throughout its burial history. This suggests that sequences of even greater antiquity will be recovered from closed systems buried in even colder sites (e.g. high altitude and/or latitude).

"25 sequences were found with an ALC (Average Local Confidence) score {greater than or equal to} 80%. This is the most stringent threshold that can be applied" If this was only from the de novo search, please include a table listing the 25 de novo peptides. It may be helpful to indicate if they match the database search as well.

We now include a detailed table with all peptide sequences *matched* to SCA-1, including the score for each peptide, therefore we replaced the statement as follows: “Eleven unique peptide sequences were found to match the sequence of struthiocalcin-1 (Table 1), all containing the typical Asp-rich motif “DDDD” (Figures 3, 4), which had been shown in our previous paper to be the mineral-binding peptide belonging to the sequence of struthiocalcin-1 (SCA-1).”

"individual annotated tandem mass spectra are shown in Figure 3 and Figure 3—figure supplement 1, Figure 3—figure supplement 2, Figure 3—figure supplement 3, Figure 3—figure supplement 4, Figure 3—figure supplement 5, Figure 3—figure supplement 6, Figure 3—figure supplement 7, Figure 3—figure supplement 8, Figure 3—figure supplement 9, Figure 3—figure supplement 10, Figure 3—figure supplement 11" Should it be Figure 4? Also, please list each peptide sequence here as well for ease of connecting to the supplemental figures.

The numbering has been kept as in the original, as figure 3 shows one of the tandem mass spectra. We have now added peptide sequences in parenthesis within the text.

Figure 4: Please include a table in the main manuscript with the 12 peptides. I appreciate that they are annotated in the supplement, but there is no reason to not include them. Alternatively, put the aligned peptides in figure 4 in place of the gray circles.

We have now added a table with the peptide sequences and their scores.

I don't know if it is possible with the software used, but please try and make this figure colorblind accessible. Red and green together are not great. Please also include details on how this figure was produced.

We have redrawn the figure, which now is a simple histogram. We hope this has improved its readability.

"demineralised in just enough cold weak hydrochloric acid" Please quantify how much acid this is.

This information was added: “The acid was added in 200 µl increments until it stopped reacting, for a total volume of 1400 µl 0.6 M HCl.”

Reviewer #2 (Recommendations for the authors):The weakest part of this manuscript is the inclusion of thermal age data, which is a metric I do not find reliable or informative in its own right, but moreover, its inclusion does not seem to add anything of value to the MS data that is presented. The thermal age estimates at the end of this paper constitute using multiple assumptions about ancient temperature and humidity values and their variability (e.g., from a "compilation of proxies") to return what the authors themselves describe as a "very crude" estimate that is nearly identical to the specimen's actual age. For what purpose? Whether "thermal age" as a metric is meant to be a proxy for determining the likelihood of the persistence of peptides (i.e., to assess that a fossil is good for analyses because it is likely less degraded than its age suggests) or to use for authentication of ancient peptides (i.e., to show that very old fossils are in fact 'thermally younger' than their geologic age because they've undergone less heating and therefore the preservation of proteins in them beyond the hypothetical limit of preservation is not problematic) both this paper and the earlier paper giving a thermal age of 16 Ma to the 3.8 Ma Laetoli eggshell show that neither usage is valid.

The thermal age was included to provide a comparative index; we have not used it to provide numerical estimates of age or as a way to make our data look “better” – in this instance the thermal age actually made our data less impressive because, coming from a colder environment, it is thermally younger than our earlier published findings. As explained above, we do however maintain that the combined effect of temperature and time cannot be ignored, indeed they must be considered. In the absence of better models for temperature variations over time an attempt of normalization, such as that provided by thermal age, provides the only sensible way for comparing samples from widely different geographic regions – as many other authors studying amino acid racemization and other forms of protein diagenesis have convincingly shown in the past.

In fact, if thermal age were to be used in this paper as it has been used in the past, it would suggest that the peptides the authors report here represent contamination. Of course, I think that would be absurd, and that the annotated spectra the authors produced in a sterile environment, with no trace of contamination in the blanks, confirm the authenticity of their peptides. However, both things can't be true. If the peptides are real--and I think they are--then thermal age is not a useful metric.

We thank the reviewer for the positive comments on our data. We however respectfully point out that the reality of the peptide and their “younger” thermal age compared to the African OES are two sides of the same coin and do not contradict each other.

The first side of the coin concerns the authenticity of the peptides recovered: we obtained these from a closed system environment – the intracrystalline fraction within ostrich eggshell; we checked for the absence of contamination. The mechanism of preservation was put forward in 2016 and is supported experimentally and independently here, by the fact that both the Laeotoli and Chines samples yielded the same peptides.The second side of the coin concerns the way proteins "age": certainly the absolute age of the specimens is crucial, but the local environment is equally important, because diagenesis is a complex network of chemical reactions, which are by definition influenced by temperature. The Chinese samples were preserved in a colder environment than the Laetoli OES, hence the rates of degradation (particularly, peptide bond hydrolysis) must have been slower than the rates of degradation at Laetoli. This is perfectly in line with the fact that peptides were recovered from this older specimen.

I strongly recommend the authors simply report their MS data, which I think are extremely impactful and worthy of publication on their own.Technical Comments:1. It is unclear to me how the authors came to the specific age range of 6.5 Ma to 9 Ma. The text is clear that 6.5 Ma is the minimum age estimate for their recovery site, and they also state that previously reported ostrich fossils were dated between 6.54-7.18 Ma, and that Platybelodon had been dated from lower in the Liushu at 11.1 – 12.5 Ma. So where does the upper bound estimate come from? Normally I wouldn't nitpick this point, but as the authors claim to have the oldest recovered peptides, the rationale for that upper bound is important and should be justified in the text. Otherwise, the authors should drop the range and stick with the minimum estimate of 6.5 Ma.

We have changed the range to the minimum age of 6.5 Ma.

2. Figure 4 is visually difficult to interpret. I strongly suggest that you change this figure to depict the letters signifying the residues as equally sized (to make them readable) and place proportionately sized bars above them (like a column chart) to show their relative coverage.

We have modified the figure following the Reviewer’s suggestions

3. Materials and methods: What is "just enough" HCl? Please give a volume.

This has been amended following both reviewers’ comment

4. It's unclear to me why the authors chose to use a fragment error tolerance of 0.5 Da, given that 0.05 Da or less is standard for ancient protein LC-MS/MS data, and that data from the Q-Exactive is certainly high enough resolution to set a higher tolerance. Additionally, the annotated peptides all appear to be within the 0.05 Da range. Is this a typo? If so, please correct it. If not, I'm curious about how a low fragment error tolerance coupled with a large search space (since it's a no-digest search) impacts the FDR rates. Please report the %FDR represented by a peptide score of −10lgP {greater than or equal to} 30 for the ostrich data.

We thank the reviewer for spotting this – it was a typo, which we have corrected. Following the Reviewers’ comments, we now report peptide matches to SCA-1 sequence rather than de novo sequences; the thresholds for peptide and protein identification were set as follows: peptide score −10lgP ≥ 15, protein score −10lgP ≥ 20, de novo sequences scores (ALC%) ≥ 80. While the FDR values are high – due to the low number of peptides, a peptide score of −10lgP ≥ 15 is standard and appears adequate here (see Author response image 1). Most peptides shown in table 1 yielded scores > 20.

**Author response image 1. sa2fig1:** Scatterplot of PEAKS peptide score versus precursor mass error.

References

Crisp M, Demarchi B, Collins M, Morgan-Williams M, Pilgrim E, Penkman K. 2013. Isolation of the intra-crystalline proteins and kinetic studies in Struthio camelus (ostrich) eggshell for amino acid geochronology. Quat Geochronol 16:110–128.

Demarchi B, Collins MJ, Tomiak PJ, Davies BJ, Penkman KEH. 2013. Intra-crystalline protein diagenesis (IcPD) in Patella vulgata. Part II: Breakdown and temperature sensitivity. Quat Geochronol 16:158–172.

Hendy EJ, Tomiak PJ, Collins MJ, Hellstrom J, Tudhope AW, Lough JM, Penkman KEH. 2012.

Assessing amino acid racemization variability in coral intra-crystalline protein for geochronological applications. Geochim Cosmochim Acta 86:338–353.

Johnson BJ, Miller GH, Fogel ML, Beaumont PB. 1997. The determination of late Quaternary paleoenvironments at Equus Cave, South Africa, using stable isotopes and amino acid racemization in ostrich eggshell. Palaeogeogr Palaeoclimatol Palaeoecol 136:121–137.

Kaufman DS. 2006. Temperature sensitivity of aspartic and glutamic acid racemization in the foraminifera Pulleniatina. Quat Geochronol 1:188–207.

Kaufman DS. 2003. Amino acid paleothermometry of Quaternary ostracodes from the Bonneville Basin, Utah. Quat Sci Rev 22:899–914.

Schroeder RA, Bada JL. 1976. A review of the geochemical applications of the amino acid racemization reaction. Earth-Sci Rev 12:347–391.

Wehmiller JF. 2013. United States Quaternary coastal sequences and molluscan racemization geochronology – What have they meant for each other over the past 45 years? Quat Geochronol 16:3–20.

Wehmiller JF. 1977. Amino acid studies of the Del Mar, California, midden site: Apparent rate constants, ground temperature models, and chronological implications. Earth Planet Sci Lett 37:184–196.

Wehmiller JF, Belknap DF. 1978. Alternative Kinetic Models for the Interpretation of Amino Acid Enantiomeric ratios in Pleistocene Mollusks: Examples from California, Washington, and Florida. Quat Res 9:330–348.

[Editors' note: further revisions were suggested prior to acceptance, as described below.]

Specifically, while you have addressed most of the reviewers' concerns, the maintained position regarding the relevance of the thermal age clashes with the strong recommendations by both reviewers (a position the reviewing editor agrees with).The geological age of the specimens should always be used as the true age, given that the thermal age is sensitive to many factors and has been repeatedly shown to artificially inflate the true age of paleontological remains. In lines 36-43 of the revised manuscript, the thermal age estimate for the Laetoli (Tanzania) specimen is invoked, despite the very large discrepancy of >10 Ma between the true age (3.6 Ma) and the thermal age (~16 Ma) of the Tanzanian sample.True temperature affects protein preservation and there are several works that have tried to quantify the extent of temperature-induced protein degradation. However, (a) there are several unknowns (estimates) that affect such assessments (e.g., past temperatures) , (b) more samples and contexts are required for a better understanding of these temperature-protein diagenesis relationships, and (c) regardless of the level of confidence of the estimated/modelled thermal age, its relevance for the current paper on ancient proteins from geologically dated OES from Linxia Basin in China is unclear (see also the disconnect in Lines 171-173).If the thermal age data are omitted altogether, or limited to a very brief and cautious mention in results/discussion, then we would likely be ready to move forward favorably with the paper. But this remaining point is considered essential.

Following the request to remove all mentions of thermal age and/or thermal age estimates, we amended the text as follows:

1) Lines 41-43. This sentence was deleted entirely: “By modelling the combined effect of temperature and time on protein diagenesis in a closed system, the peptides recovered from the Laetoli OES were estimated to be equivalent to ~16 Ma at a constant 10°C”

2) Lines 170-173. The sentence: “The absolute age of the Chinese sample is greater than that of the African material (Demarchi et al., 2016), but it is estimated to have experienced lower temperatures throughout its burial history. This suggests that sequences of even greater antiquity will be recovered from closed systems buried in even colder sites (e.g. high altitude and/or latitude)“ was reworded more cautiously to “The absolute age of the Chinese sample is greater than that of the African material (Demarchi et al., 2016), but, given the latitudinal differences between the two sites, it is likely to have experienced lower temperatures throughout its burial history. This suggests that sequences of even greater antiquity may be recovered from biominerals harbouring a closed system of proteins, particularly from sites in cold environments (e.g. high altitude and/or latitude).”